# Does Climate Change Affect Rapeseed Production in Exporting and Importing Countries? Evidence from Market Dynamics Syntheses

Arifa Jannat [1,2] , Yuki Ishikawa-Ishiwata [3] and Jun Furuya [4,*]

1 Graduate School of Life and Environmental Sciences, University of Tsukuba, 1-1-1 Tennodai, Tsukuba 305-8572, Ibaraki, Japan; s1936023@s.tsukuba.ac.jp

2 Institute of Agribusiness and Development Studies, Bangladesh Agricultural University, Mymensingh 2202, Bangladesh

3 Global and Local Environment Co-Creation Institute (GLEC), Ibaraki University, 2-1-1 Bunkyo, Mito 310-8512, Ibaraki, Japan; yuki.ishikawa.ga@vc.ibaraki.ac.jp

4 Social Sciences Division, Japan International Research Center for Agricultural Sciences, 1-1 Owashi, Tsukuba 305-8686, Ibaraki, Japan

* Correspondence: furuya@affrc.go.jp; Tel.: +81-29-838-6304

**Abstract:** Globally, non-cereal crops such as vegetable oils and their associated products will surpass cereals in demand by 2050, according to the World Bank (WB). Despite being considered an energy-efficient food crop, the production and supply capability of rapeseed is mostly influenced by climate conditions. Aiming in this context, the study explored how temperature and rainfall patterns influence rapeseed production, as well as how rapeseed prices in major trading countries may influence production and consumption patterns in developing countries. To do this, a supply and demand model approach has been employed for major exporting (Canada) and importing countries, i.e., China, the United States (U.S.) along with Bangladesh, a developing nation. The baseline study period was considered from 1991 to 2018, and simulations were performed up to 2040. The findings revealed that the most important effect on rapeseed yield is directly related to changes in temperature, which are positively related to the growing season but negatively related to the maturity stages of rapeseed in all studied countries. Rapeseed exports from Canada are expected to rise by 2040, while imports from China and the U.S. will rise simultaneously. In Canada, the per capita consumption of rapeseed oil is expected to increase from 16 to 24 kg per year between 2019 and 2040. Over the projection period, oil per capita consumption has steadily increased in China, the U.S., and Bangladesh. The relative demand for rapeseed oil is projected to increase by 2060, according to representative concentration pathways (RCPs). Therefore, it is necessary to determine market prices considering the probable climatic effect and increasing market demand for rapeseed to sustain the international market access of trading nations.

**Keywords:** supply and demand model; oilseed rape; RCPs; SSPs; climate change; price linkage

## 1. Introduction

Climate change affects global land area and agricultural production in a variety of ways, including differences in annual rainfall, average temperature, heat waves, $CO_2$ emissions, etc. [1,2]. Among the most important food crops are cereal grains, including wheat, maize, and paddy, which are staple crops for most of the world's population. The World Bank Group research has found that by 2050, the demand for non-cereal crops such as vegetable oils, sugar, roots, and tubers, as well as pulses, will increase by 79%, 56%, 48%, and 41%, respectively, compared to cereals (32%) [3]. The FAO has found that cereal foods alone cannot meet the entire nutritional needs of humans globally [4,5]. Non-cereal foods must also be considered for a well-balanced diet. Even though food supplies have

increased in recent decades, chronic nutritional deficiency has become widespread due to a declining trend in daily calorie consumption or an imbalanced diet. Consumption habits and improper use of natural resources destroy environmental sustainability in the long run. It is important to strike a balance between ensuring increased productivity of vegetable oils in meeting global demand as well as ensuring environmental sustainability. The United Nations (UN) 2030 agenda on Sustainable Development Goals (SDGs) aims to promote the goals (SDGs 2, 7, 12, and 13) of sustainable cultivation of oilseeds and the production and processing of vegetable oils [6].

Oilseeds are fourth among the essential food commodities, behind cereals, vegetables and fruits, and account for 213 million hectares (ha) of arable land [7]. Due to population growth, dietary diversity, affluence across the globe, and a need for more sustainable bioproducts, oil crop utilization and demand have continuously increased over the years [8]. Apart from its potential as a biofuel, vegetable oil also serves as a sustainable source of energy [9].

Rapeseed (*Brassica napus*, *B. rapa*, and *B. juncea*) is the world's third most important oilseed. They include crude and refined fractions of canola, rape, colza or mustard oil. It is primarily grown for its oil but is also used for edible and industrial purposes. Rapeseed is an annual herb from relatively cool and humid temperate climates. Rapeseed, which contains more than 40% oil, is more profitable than soybeans, which contain only 18% oil [10]. It provides good soil cover over winter to prevent soil erosion, produces large amounts of biomass, suppresses weeds, and can improve soil tilth with its root system [10]. However, although rapeseed is regarded as an energy-smart food crop, climate conditions can influence its productivity and supply capability, similarly to other crops. To meet future oilseed demand, a potential approach is the development of new alternative crops of oilseed rape that are naturally adapted to more xeric conditions [11]. Global demand for edible oil products has also increased as a result of growing populations, rising affluence, rapid urbanization, and changing dietary preferences [12].

To meet the Food and Agriculture Organization (FAO) projections for food, fuel, and industrial demand, the global production of vegetable oils is expected to double by 2050 [13]. The leading global producers of rapeseed/canola are Canada (21 million MT), the EU (19 million MT), and China (13 million MT), with an annual production of over 70 million MT [14,15]. The major importers are China, the U.S., the EU, etc. [14]. Rapeseed oil has 59% of the total global biodiesel raw material sources, followed by soybean (25%), palm oil (10%), sunflower oil (5%), and other (1%) [16]. Although rapeseed is the most dominant oilseed crop in Bangladesh, a significant number of oilseeds are imported from Canada and other exporting countries every year. However, in most cases, oil-importing developing countries (i.e., India, Bangladesh, Pakistan) with large population densities are fully dependent on major trading countries such as Canada, Germany, and other countries. Since the Bangladesh rapeseed market fully relies on importing oil and oil products, its inflation is also largely determined by international prices, even though domestic production significantly impacts output and price stability.

Some studies have already been conducted regarding production risk under climate conditions between two rapeseed varieties [17,18]. Besides this, a comprehensive evaluation of climate change impacts on rapeseed production under different levels of global warming has been conducted [19]. Furthermore, climate effects on suitable potential rapeseed areas [11] and the impact of heat and drought stresses on rapeseed production practices have also been carried out [20]. As stated by Tabassum (2015), not only climate issues related to market dynamics but also the plummeting price of oil are currently the most sensational energy stories in the world. The dynamics were rapidly shifting and wreaking havoc on oil-producing countries' economies [21].

Therefore, it is essential to analyze the possible scenarios for rapeseed market dynamics in major exporting and importing countries. In this connection, the supply and demand model approach is advantageous over other econometric analyses. Different researchers have applied supply and demand models to the market dynamics of rice [22–24],

soybeans [25], and potatoes [26]. The supply and demand model can analyze changes in yield and planted area independently by incorporating climate parameters and considering supply responses and demand changes in response to market variations by equating supply and demand [27]. Some research has already been conducted regarding environmental life cycle assessment of rapeseed [28], environmental impacts of rapeseed and rapeseed oil [29], environmental and economic assessment of rapeseed [30], and assessment of the effects of GHGs on rapeseed cultivation [31].

For climate impact assessment, production estimation, and simulations for rapeseed, several crop models have been developed, including APSIM (Agricultural Production Systems Simulator), CSM-CROPGRO-Canola, EPIC (Environmental Policy Integrated Climate), Agro-Ecological Zone (AEZ), and Crop Simulation Models [32–35]. Existing simulations of rapeseed production potential have shown a lack of attention to market dynamics of rapeseed under historical and future climate change. Our study made an effort to combine this supply and demand mechanism to synthesize the market dynamics of rapeseed for major trading countries, considering changes in temperature and rainfall patterns.

In this connection, it is very important to address the effect of global market dynamics of oilseed rape under different climate conditions and how rapeseed market prices in the major trading countries affect production and consumption patterns in a developing country, such as Bangladesh. Keeping in mind the above issues, this paper simulates the impact of climate change on rapeseed production potential in the major exporting and importing countries. Moreover, this study further evaluated the probable scenarios of the rapeseed market in Bangladesh with the changing equilibrium price of trading countries. Materials and methods with data and analytical techniques are presented in Section 2. The Section 3 focuses on the results and findings of the study. Following this, the discussion is covered in Section 4. Lastly, Section 5 reports the conclusion and policy implications of the paper.

## 2. Materials and Methods

### 2.1. Study Area

In 2019, Canada exported 3.23 billion U.S. dollars (USD) in rapeseed, making it the world's largest exporter of rapeseed. In the same year, rapeseed was the 20th most exported product in Canada. The main destinations of rapeseed exports from Canada are Japan (828 million USD), China (626 million USD), and Mexico (393 million USD) [36]. In 2019, the top importers of rapeseed oil were the US (1.59 billion USD), China (1.15 billion USD), and the Netherlands (736 million USD). Considering the trade value, this study chose Canada as the major exporting country and China and the U.S. as the major importing countries. These three countries are very large, and the climate patterns are also different in each area. Therefore, we considered the climatic features of the major rapeseed growing areas, irrespective of the entire country, to assess more specific climate change impacts on production. The study area map with major growing areas of rapeseed in Canada, China, and the U.S. was created in the ArcGIS environment (Figure A1). Basic layers were projected in the WGS84 using ArcGIS 10.8.1® software. More specifically, to determine the market dynamics of rapeseed under the climate variability of a developing country, we chose Bangladesh, where rapeseed is the main oilseed crop (Figure A2).

### 2.2. Data Sources and Acquisition

#### 2.2.1. Baseline Data

Different studies have explored whether the two most significant abiotic factors limiting crop productivity around the world are temperature and rainfall [37–39]. Based on major rapeseed growing regions in all countries and data availability, the basic assessment was conducted from 1991–2018. Our study considered temperature and rainfall as the most influential climate variables, and data were extracted from different climate stations at the National Centers for Environmental Information [40]. For much of the growing season, a field-grown crop is exposed to heat stress [41]. The large differences in yield between

areas with cooler versus warmer temperatures show that crop yields are reduced due to high-temperature stress [42].

The temperature and rainfall data were gathered from the Bangladesh Metrological Department (BMD) for Bangladesh. Then, the monthly mean temperature and rainfall data were calculated for each study area, followed by the rapeseed cropping calendar. However, the simulation period was extended from the historical period to 2040. Before analyzing the supply and demand model for rapeseed, it is important to know the life cycle of rapeseed production. In this regard, the rapeseed crop calendar for major trading countries, including Bangladesh, is presented in Figure A3.

Time-series data for the historical yield and planted area of rapeseed were collected from the Food and Agricultural Organization's statistical database, FAO–STAT (2021) [43]. The market price of rapeseed in Canada, China, and the U.S. was assumed to be the farm price (*FP*); world price (*WP*), imports (*IMP*), exports (*EXP*), food demand, seed, loss, and biofuels (other usages), and other macroeconomic indicators, such as real gross domestic product (*GDP*), GDP deflator (*GDPD*), population (*POP*), and exchange rate (*EXR*), were collected from the FAOSTAT (2021) database [40]. Data related to the consumer price index (*CPI*) and crude oil world price (*WPcru*) were collected from the World Bank [44].

### 2.2.2. Projected Data

The projected period ran from 2019 until 2060 for the market dynamics of rapeseed. In the multimodel ensemble (MME), climate sensitivity (CS) was created by employing different models of structures and resolutions in response to surface temperature and $CO_2$ concentration. To acknowledge parametric uncertainty, the Model for Interdisciplinary Research on Climate (MIROC5) has been developed as a perturbed physics ensemble (PPE) that links atmosphere–ocean general circulation models (CGCM) [45]. MIROC5, the General Circulation Model (GCM) of the University of Tokyo, the National Institute for Environmental Studies (NIES), and the Japan Agency for Marine-Earth Science and Technology (JAMSTEC) were used as the sources of future climate data in the simulations. To avoid the substantial limitations of previous PPE studies (ASGCM), large radiation imbalance at the top of the atmosphere, and climate drifts, MIROC5 developed a method of controlling TOA imbalance in the CGCM PPE without flux corrections.

Thus, MIROC5 has been distinguished as producing a relatively accurate climate scenario [46]. The Intergovernmental Panel on Climate Change (IPCC) AR5 developed four representative concentration pathway (RCP) scenarios (RCP 2.6, RCP 4.5, RCP 6.0, and RCP 8.5). They can be delineated based on radiative forcing and the direction of change. A set of five shared socioeconomic pathways (SSPs) (SSP1–SSP5) that explain the microscale conditions of human and natural resources was considered based on O'Neill et al. (2017) [47].

In all, 20 possible scenarios (four RCPs × five SSPs) exist in the RCP–SSP combination. The SSPs contain narratives for future demographics, economy and lifestyle developments, policies and institutions, technology, and the environment and natural resources [47]. However, all scenarios for each country are different because of variations in the GDP growth rate, population pressure, and climate change adaptation and mitigation challenges. In addition, each scenario had its own fluctuation of climate variables, which would cause the fluctuation of rapeseed production.

The projected GDP and population under the SSPs of the 5th Assessment Report (AR5) of the IPCC were incorporated into climate scenarios to predict the food situation and price instability in the future under different climate conditions. Furthermore, the SSPs comprise quantitative projections of population and GDP at the country level [48,49]. This study focuses on all SSPs based on their challenges and strategies for the studied countries. With all RCPs, SSP1, SSP2, SSP3, and SSP5 were selected for Canada and the U.S. according to the challenges, while SSP2, SSP3, and SSP4 were considered for China and Bangladesh.

### 2.3. Analytical Techniques

Supply and Demand Mechanism

To generate the outlook on the variation in the supply and market price of rapeseed for Canada, China, the U.S., and Bangladesh under different climate conditions, the basic structure of the model was developed and modified by following Ishikawa-Ishiwata and Furuya (2021), Furuya et al. (2010) and Koizumi and Ohga (2008) [25,27,50].

The primary model of this research is described as follows.

Yield Function:

$$Y_{RY,k,t} = \beta_{Y,0} + \beta_{Y,1} \, TEMP_{k,i,t} + \varepsilon_{Y,\,t} \tag{1}$$

where $Y_{RY,kt}$ is the yield of rapeseed (MT ha$^{-1}$), $TEMP_{i,t}$ denotes the monthly average temperature, $\beta_{Y,0}$, and $\beta_{Y,1}$ are the parameters, and $\varepsilon_{Y,t}$ is the error term. The subscripts $k$ and $t$ denote the country (4 countries) and the year, respectively.

Area Function:

$$A_{RA,k,t} = \beta_{A,0} + \beta_{A,1}A_{k,\,(t-1)} + \beta_{A,2}FPd_{k,(t-1)} + \beta_{A,3} \, Rain_{k,i,(t-1)} + \varepsilon_{A,\,t} \tag{2}$$

where $A_{RA,k,t}$ is the rapeseed harvested area for time t, and $A_{k,(t-1)}$ is the one-year lagged area. $FPd_{k,(t-1)} = FP_{k,(t-1)}/(CPI_{k,(t-1)} \times 100)$ (Local currency MT$^{-1}$) is the one-year lagged farm price of rapeseed deflated by the lagged consumption price index $CPI_{t-1}/100$. $CPI$ is the consumer price index (2015 = 100). $Rain_{k,i,(t-1)}$ is the one-year lagged average rainfall. $\beta_{A,0}$, $\beta_{A,1}$ and $\beta_{A,3}$ are the parameters, and $\varepsilon_{A,t}$ is the error term.

Total Production:

$$TQ_{R,k,\,t} = Y_{RY,k,t} \times A_{RA,k,t} \tag{3}$$

where $TQ_{R,k,\,t}$ denotes the total production for rapeseed.

Export function:

The export function for the rapeseed exporting country is as follows:

$$EXP_{R,k,t} = \beta_{EXP,0} + \beta_{EXP,1}TQ_{k,t} + \beta_{EXpP,2}FPd_{k,t} + \varepsilon_{EXP,\,t} \tag{4}$$

Import function:

The import function for rapeseed importing countries is

$$IMP_{R,k,t} = \beta_{IMP,0} + \beta_{IMP,1}TQ_{R,k,t} + \beta_{IMP,2}WPd_{R,k,t} + \varepsilon_{IMP,\,t} \tag{5}$$

where $IMP_{R,k,\,t}$ is the quantity of imports (MT) for importing countries, $TQ_{k,t}$ is the total domestic production (MT) of rapeseed, and $WPd_{R,k,\,t} = WP_{k,t} \times EXR_{k,t}/(CPI_{k,t}/100)$ is defined as the world price of rapeseed (USD), where $EXR_{k,t}$ is the exchange rate (local currency/USD), $\beta_{IMP,\,0}$ and $\beta_{IMP,1}$ are the parameters, and $\varepsilon_{IMP,t}$ is the error term.

Stock change function:

$$STC_{R,k,t} = \beta_{STC,0} + \beta_{STC,1}(TQ_{R,k,t} - TQ_{R,k,(t-1)}) + \varepsilon_{STC,\,t} \tag{6}$$

where $STC_{R,k,t}$ is the quantity of stock change (MT), i.e., ending stock minus beginning stock.

Total supply processing identity:

$$PROC_{R,k,t} = TQ_{R,k,t} + IMP_{R,k,t} - EXP_{R,k,t} - STC_{R,k,t} - FEED_{R,k,t} - FOOD_{R,k,t} - LOSS_{R,k,t} - OBD_{R,k,t} \tag{7}$$

where $PROC_{R,k,t}$ is the net supply (MT) for each country. $FEED_{R,k,t}$, $FOOD_{R,k,t}$, $LOSS_{R,k,t}$ and $OBD_{k,t}$ are the feed quantity, food, loss during the process and biodiesel usages of rapeseed, respectively (MT).

Oil processing identity:

$$OilQ_{R,k,t} = PROC_{\,k,t} - CakeQ_{R,k,t} \tag{8}$$

where $OilQ_{R,k,t}$ denotes the oil processing identity of rapeseed. $CakeQ_{R,k,t}$ is the quantity of rapeseed cake production (MT) used by each country.

Oil export function:

The export function for the rapeseed oil exporting country is as follows:

$$OEXP_{R,k,t} = \beta_{oilEXP,0} + \beta_{EXP,1}OilQ_{R,k,t} + \beta_{oilEXP,2}FPd_{k,t} + \varepsilon_{oilEXP,\,t} \tag{9}$$

where $OEXP_{R,k,t}$ is the quantity of rapeseed oil exports (MT). $\beta_{oilEXP,1}$ and $\beta_{oilEXP,2}$ are the parameters, and $\varepsilon_{oilEXP,t}$ is the error term.

Oil import function:

The import function for rapeseed oil-importing countries is

$$OIMP_{R,k,t} = \beta_{oilIMP,0} + \beta_{oilIMP,1}OilQ_{R,k,t} + \beta_{oilIMP,2}WPd_{R,k,t} + \varepsilon_{oilIMP,\,t} \tag{10}$$

where $OIMP_{R,k,t}$ is the quantity of rapeseed oil imports (MT). $\beta_{oilIMP,\,0}$, $\beta_{oilIMP,1}$ and $\beta_{oilIMP,\,2}$ are the parameters, and $\varepsilon_{oilIMP,t}$ is the error term.

Price linkage function:

Price linkage function for importing countries is-

$$OilWP_{R,t} = \beta_{WP,0} + \beta_{WP,1}FP_{R,k,t} + \varepsilon_{WP,\,t} \tag{11}$$

where $OilWP_t$ is the rapeseed oil world price (USD). $\beta_{WP,0}$ and $\beta_{WP,1}$ are the parameters. $\varepsilon_{WP,\,t}$ is the error term.

Oil biodiesel function:

The oil biodiesel function for rapeseed is as follows:

$$OBD_{R,k,t} = \beta_{oilBD,0} + \beta_{oilBD,1}WPdcru_t + \beta_{oilBD,2}WPdsoy_t + \varepsilon_{oilBD,\,t} \tag{12}$$

where $OBD_{k,t}$ is the quantity of rapeseed oil used as biodiesel (MT). $WPdcru_t = WPcru_t \times EXR_t/(CPI_t/100)$ is defined as the world crude oil price (USD), where $EXR_t$ is the exchange rate (local currency USD$^{-1}$). The world price of crude oil is represented by $WPcru_t$, used as a proxy price of biodiesel oil, which is the average spot price of Brent, Dubai, and West Texas Intermediate, all of which are calculated equally. $WPdsoy_t$ is the world price of soybeans (USD MT$^{-1}$). $\beta_{oilBD,1}$ and $\beta_{oilBD,2}$ are the parameters, and $\varepsilon_{oilBD,t}$ is the error term.

Biodiesel price linkage function:

The biodiesel price linkage function is as follows:

$$FP_{R,t} = \beta_{WP,0} + \beta_{WP,1}WPcru_{R,k,t} + \varepsilon_{WP,\,t} \tag{13}$$

where $\beta_{WP,0}$ and $\beta_{WP,1}$ are the parameters. $\varepsilon_{WP,\,t}$ is the error term.

Oil supply identity:

$$OilQS_{R,k,t} = OilQ_{R,k,t} + OIMP_{R,k,t} - OEXP_{R,k,t} - OSTC_{R,k,t} - OBD_{R,k,t} \tag{14}$$

where $OilQS_{R,\,k,t}$, $OEXP_{R,k,t}$ and $OSTC_{R,k,t}$ denote the total edible oil supply, oil exports, and oil stock change of rapeseed, respectively (MT).

Per capita oil food consumption:

$$Food_{oil,\,R,k,t} = \beta_{Food,0} + \beta_{Food,1}FPd_{R,k,t} + +\beta_{Food,2}\,GDPPCd_{k,t} + \beta_{Food,3}FPdsub_{k,t} + \varepsilon_{Food,\,t} \tag{15}$$

where $Food_{oil,\,R,k,t}$ denotes the per capita oil consumption of rapeseed, which is determined by dividing total oil food consumption by total population, and $GDPPCd_{k,t}$ ($rGDP_{k,t}/POP_{k,t}$) is derived from the real GDP for each country, which is divided by population ($POP_t$). Here, the real $rGDP_t$ is the GDP that is transformed into constant international dollars (base 2015) followed by purchasing power parity (PPP) rates. $FPdsub_{k,t}$

is the real farm price of substitutes of rapeseed oil, i.e., soybean oil and palm oil, which can be defined as $FPsub_{k,t}/(CPI_{k,t}/100)$.

Oil demand identity:

$$OilQD_{R,k,t} = Food_{oil,\,R,k,t} \times POP_{k,t} \tag{16}$$

where $OilQD_{,k,t}$ is the total edible oil demand.

Market equilibrium identity:

$$OilQS_{R,k,\,t} = OilQD_{R,\,k,t} \tag{17}$$

To understand the market dynamics mechanism, we present a Price and Quantity (PQ) space diagram for exporting and importing countries, incorporating climate variables (Figures 1 and 2). It represents the interaction between the farm price and quantity of rapeseed with associated parameters. Assuming other things remain the same, we assume that rainfall has a positive effect on planted area allocation, while the temperature negatively affects the rapeseed yield. Then, the total production with imports was adjusted to the domestic supply by subtracting exports, stock changes, losses, and other usages of rapeseed. After being processed, the rapeseed can be separated into oil and cake, where the oil is used for food and biodiesel. World rapeseed oil price influences domestic oil supply. Rapeseed oil demand is primarily determined by a country's per capita GDP and population. After estimating the total rapeseed oil supply and demand, in the final step, the procedures were also executed by an equilibrator to find the expected point of convergence, which indicates the domestic market price in a spreadsheet. Shifts in market prices for rapeseed are influenced by supply and demand interactions. The iteration of price adjustments can be conducted repetitively in the following procedures.

In that equation, *DF* is the damping factor, which can be measured as a constant number:

$OilQS_{R,k,t} > OilQD_{R,k,t}$ when $ADV_t$ became negative and $FP_{R,k,t}$ decreased;

$OilQS_{R,k,t} < OilQD_{R,k,t}$ when $ADV_t$ became positive and $FP_{R,k,t}$ increased;

$$\text{Adjusted value } (ADV_t) = (OilQS_{R,k,t} - OilQD_{R,k,t}) \times (-DF) \tag{18}$$

The iteration process was terminated when $ADV_t \approx 0$.

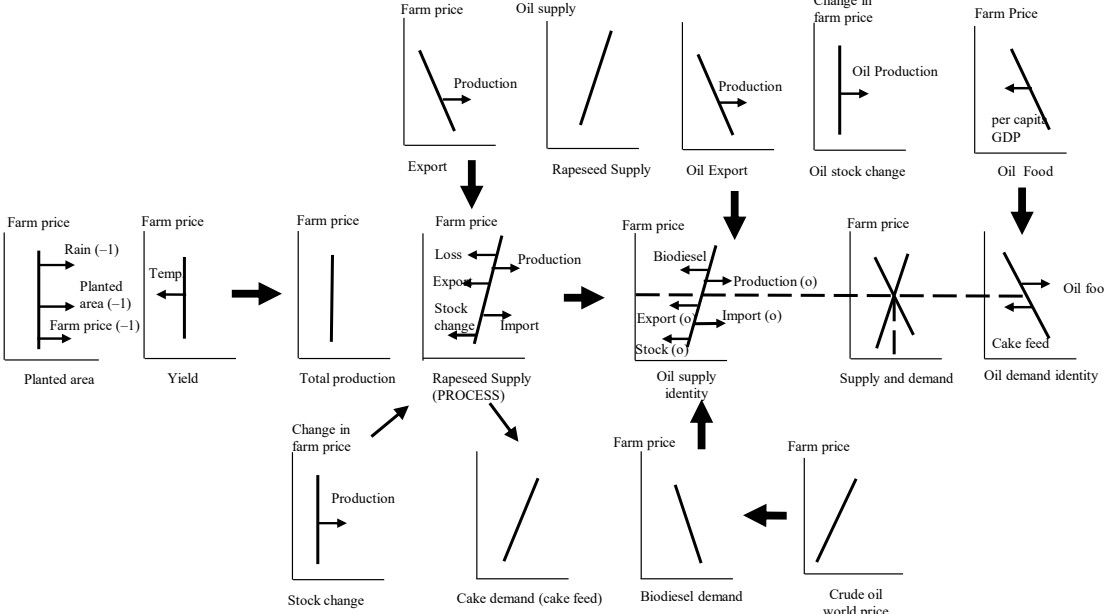

**Figure 1.** PQ space of the rapeseed econometric model in exporting country.

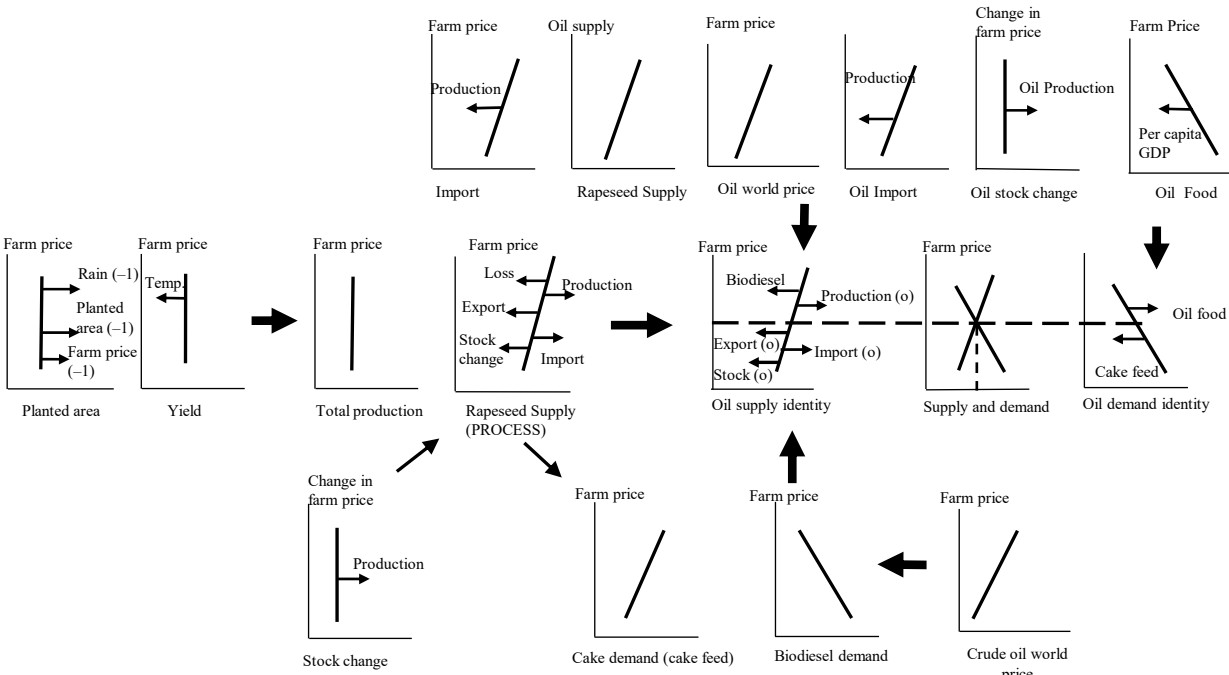

**Figure 2.** PQ space of the rapeseed econometric model in importing countries. Source: Authors' own creation.

The mentioned statistical analyses were conducted to apply the above econometric model of supply and demand for rapeseed. After the descriptive statistics, the correlation coefficient was measured to identify the relationship between the trading countries' dependent (yield, area) and independent (temperature and rainfall) variables. Augmented Dickey–Fuller (ADF) and Phillips–Perron (PP) tests were performed to assess all variables' stationarity [51,52]. The results indicated that most variables had unit-roots (Tables 1 and 2), and almost all climate variables were stationary (Table A1). If the variables had unit-roots, then Johansen cointegration tests were used in the next step to identify the existence of a long-run equilibrium relationship between the variables, although the variables have unit roots [53]. The first-order difference technique was used to make all variables stationary to ensure uniform estimation methods for each region. The estimation technique was the ordinary least-squares (OLS) method.

**Table 1.** Unit root tests for Canada and China.

| Variables | Levels | | 1st Differences | | Variables | Levels | | 1st Differences | |
|---|---|---|---|---|---|---|---|---|---|
| | ADF | PP | ADF | PP | | ADF | PP | ADF | PP |
| | | | | | Canada | | | | |
| *A* | −2.51 | −2.48 | −4.99 *** | −8.02 *** | *OEXP* | −1.75 | −0.91 | −4.34 *** | −4.05 *** |
| *Y* | −4.52 *** | −4.50 *** | −4.57 *** | −22.39 *** | *OSTC* | −6.12 *** | −6.06 *** | −7.06 *** | −31.55 *** |
| *Q* | −2.06 | −1.92 | −6.49 *** | −7.45 *** | *OBD* | −3.53 *** | −3.61 *** | −5.27 *** | −15.07 *** |
| *FP* | −3.31 *** | −1.98 | −4.28 | −4.33 *** | *CruWP* | −1.92 | −2.05 | −3.98 *** | −7.23 *** |
| *EXP* | −2.71 | −2.73 | −3.98 *** | −7.23 *** | *GDP* | −2.75 | −2.88 | −4.23 *** | −5.34 *** |
| *STC* | −4.57 *** | −7.98 *** | −4.35 *** | −13.76 *** | *POP* | −1.84 | −0.92 | −6.49 *** | −2.75 |
| *OilQ* | −2.38 | −0.75 | −4.85 *** | −4.88 *** | - | - | - | - | - |

**Table 1.** *Cont.*

| Variables | Levels | | 1st Differences | | Variables | Levels | | 1st Differences | |
|---|---|---|---|---|---|---|---|---|---|
| | **ADF** | **PP** | **ADF** | **PP** | | **ADF** | **PP** | **ADF** | **PP** |
| | | | | China | | | | | |
| *A* | –2.31 | –1.69 | –3.00 *** | –5.71 *** | *OIMP* | –1.95 | –3.00 | –3.87 ** | –6.94 *** |
| *Y* | –3.58 * | –3.57 * | –7.98 *** | –9.82 *** | *OBD* | –1.53 | –1.61 | –4.07 *** | –8.27 *** |
| *Q* | –1.02 | –1.67 | –3.52 * | –9.65 *** | *WP* | –3.33 * | –1.91 | –4.95 *** | –4.61 *** |
| *FP* | –1.44 | –1.56 | –3.54 *** | –4.90 ** | *CruWP* | –1.92 | –2.05 | –4.61 *** | –4.58 *** |
| *IMP* | –2.29 | –2.32 | –3.51 * | –4.92 *** | *GDP* | 0.80 | 1.42 | –3.58 * | –2.38 |
| *OilQ* | 0.67 | 0.21 | –4.09 ** | –4.04 ** | *POP* | –4.83 *** | –16.49 | –5.12 | –5.26 *** |

Source: Authors' estimation based on data, 2021. Note: All the unit root tests include both a constant and a linear trend. ***, **, and * denote significance at 1%, 5%, and 10% levels, respectively.

**Table 2.** Unit root tests for the U.S. and Bangladesh.

| Variables | Level | | 1st Differences | | Variables | Level | | 1st Differences | |
|---|---|---|---|---|---|---|---|---|---|
| | **ADF** | **PP** | **ADF** | **PP** | | **ADF** | **PP** | **ADF** | **PP** |
| | | | | U.S. | | | | | |
| *A* | –2.20 | –2.58 | –4.38 ** | –11.93 *** | *OIMP* | –1.72 | –0.92 | –3.86 * | –7.85 *** |
| *Y* | –1.36 | –5.17 | –7.84 *** | –16.65 *** | *OBD* | –2.34 | –1.93 | –4.67 *** | –11.01 *** |
| *Q* | –2.09 | –1.12 | –8.65 *** | –9.79 ** | *WP* | –1.72 | –2.32 ** | –6.17 *** | –8.19 *** |
| *FP* | –1.65 | –2.18 | –7.14 *** | –8.97 *** | *CruWP* | –1.84 | –1.84 | –6.75 *** | –6.78 *** |
| *IMP* | –2.76 | –3.55 *** | –4.45 *** | –10.40 *** | *GDP* | 0.50 | 0.14 | –5.25 *** | –5.23 *** |
| *OilQ* | –0.67 | –1.12 | –8.65 *** | –9.79 *** | *POP* | –0.90 | 0.17 | –3.21 * | –6.37 *** |
| | | | | Bangladesh | | | | | |
| *A* | –3.98 ** | –1.18 | –5.87 *** | –3.31 * | *OilQ* | –2.93 * | –2.35 | –4.68 *** | –4.94 *** |
| *Y* | –2.14 | –1.98 | –5.27 *** | –9.23 *** | *WP* | –3.07 ** | –2.43 | –5.53 | –6.55 *** |
| *Q* | –3.57 * | –0.84 | –4.91 *** | –4.94 *** | *GDP* | 0.19 | 2.25 * | –3.59 * | –4.13 |
| *FP* | –1.82 | –1.80 | –4.98 *** | –4.93 *** | *POP* | –0.84 | –0.97 | –3.30 | –7.31 *** |
| *IMP* | –2.09 | –2.19 | –5.31 *** | –5.34 *** | - | - | - | - | - |

Source: Authors' calculation, 2021. Note: All the unit root tests include both a constant and a linear trend. ***, **, and * denote significance at 1%, 5%, and 10% levels, respectively.

To test whether the models contain serial correlation and heteroskedasticity issues, the Breusch–Godfrey test for autocorrelation [54,55] and the Breusch and Pagan (1979) test for heteroskedasticity were performed [56]. When serial correlation was found in the error term, an autoregressive (AR1) model was applied. The model's goodness of fit was identified with the information criteria, i.e., the Akaike information criterion (AIC), whereby the lower the values of those criteria are, the better the model specification. The F-stat was regarded as identifying the model accuracy. However, the Durbin-Watson Stat (DW Stat) shows autocorrelation in residuals. If the DW stat value is nearly two, then the model can be regarded as "autocorrelation free". We regard this as the lag of the dependent variable in the regression equation. Alternative DW statistics (Durbin h statistics) were considered if the regression model includes the lagged dependent variable. The analyses were performed by Stata Statistical Software 14.0, EViews 12th edition, and SAS 9.4.

Then, we made projections by using equilibrators under different RCPs and SSPs for Canada, China, the U.S., and Bangladesh to determine how climate change might influence the supply, demand, and price of rapeseed. By calculating the coefficient of variation (CV),

we looked at the extent of relative changes in rapeseed areas, production, consumption, and farm prices in Bangladesh under different scenarios and assumptions about climate action.

## 3. Results

### 3.1. Estimation of Functions of the Supply and Demand Model

Table 3 presents the relationships between the rapeseed yield and the area with temperature and rainfall. Different environmental conditions were observed during the rapeseed production season in our studied countries. For all countries, the growing season's temperature had a significant relationship with yield. Rapeseed is produced in cool weather conditions with an optimum temperature for growth and production of 20–21 °C [57]. Temperatures above 30 °C are detrimental to flower pollination and will shorten the pod and seed development phase to the point where yield and quality are compromised. In Canada, the average temperature in July is ideal for growing rapeseed and has a strong relationship with yield. In China and the U.S., the average temperature in November is effective for yield growth.

**Table 3.** Estimation of rapeseed yield and area function with climatic parameters.

| Parameters | | Canada | China | U.S. | Bangladesh |
|---|---|---|---|---|---|
| | | Yield | | | |
| | Jul | 0.090 * (7.76) | | | |
| | Aug | −054 * (−5.38) | | | |
| Temp. | Nov | | 0.050 *** (1.93) | 0.020 * (4.21) | |
| | Dec | | −0.019 ** (−2.43) | −0.012 ** (−3.02) | 0.039 * (5.15) |
| | Jan | | | | |
| | Feb | | | | −0.019 * (−4.76) |
| Adj. $R^2$ (DW) | | 0.67 (2.50) | 0.53 (1.99) | 0.75 (2.47) | 0.67 (2.13) |
| | | Area | | | |
| $A_{t-1}$ | | 0.378 ** (2.14) [0.37] | 0.180 ** (1.85) [0.18] | 0.364 ** (1.94) [0.34] | 0.329 ** (3.01) [0.33] |
| $FPd_{t-1}$ | | 2919.493 ** (1.77) [0.18] | 365.677 ** (2.52) [0.16] | 358.4571 (1.67) [0.22] | 0.908 *** (1.85) [0.10] |
| | $Jan_{t-1}$ | - | | 9142.189 * (4.74) | |
| | $Feb_{t-1}$ | | | | −444.063 * (−5.52) |
| Rain. | $Mar_{t-1}$ | - | | 3283.541 * (3.29) | |
| | $May_{t-1}$ | - | | −556.718 ** (−1.91) | |
| | $Jun_{t-1}$ | 7934.163 ** (1.77) | | | |
| | $Oct_{t-1}$ | | | | 65.212 ** (3.81) |
| | $Nov_{t-1}$ | | 9585.424 ** (2.3) | | |
| Adj. $R^2$ | | 0.65 | 0.86 | 0.90 | 0.77 |

Source: Authors' estimation, 2021. Note: ***, **, and *, respectively denote the levels of significance at 1, 5, and 10%. Values in () and [], respectively denote t-values and elasticity. DW denotes the Durbin–Watson test statistics value and AdjR² is adjusted R². January (Jan), February (Feb), March (Mar), April (Apr), May (May), June (Jun), July (Jul), August (Aug), October (Oct), November (Nov), December (Dec).

The growing season of rapeseed is longer in the U.S. and China than in Canada and Bangladesh. As a temperate region, the average temperature in December in Bangladesh is

20 °C, which has a positive effect on rapeseed yield. However, the average temperature becomes higher in February than in December, and can cause substantial yield losses of approximately 1.9% (Table 3). High temperatures hasten plant growth, shorten the growing season, and reduce yield potential [58]. Although high temperatures have a significant impact on crop yield, little attention is given to the impact of high temperatures combined with rain [40].

Rainfall distribution is very important, and a long rainy season with sufficient rain and cooler climatic conditions during the pod and seed development stages is very important. In general, rainfall (more than 400 mm) affects pod and seed development in most areas, but water stress is susceptible after maturity [42]. Along with the results shown in Table 3, higher farm prices encouraged farmers to expand their planted areas of rapeseed in all countries. The area function findings showed that the previous year's cultivation area positively influenced the current year's rapeseed cultivation area.

This study also found that in every crop year, rapeseed production areas increased steadily in Canada, China, the U.S., and Bangladesh. Similar findings were also observed by Elferjani and Soolanayakanahally (2018) and Begna et al. (2017) [59,60]. The elasticity of the lagged year's farm price was 0.10–0.22, indicating that rapeseed prices played a significant role in deciding the allocation of lands and that they had a positive and statistically significant effect on rapeseed producer behavior.

The fluctuations in farm prices during the simulated period for the studied countries are illustrated in Figure 3. The box plot visualization shows that the farm price fluctuated less in Canada and the U.S. than in China and Bangladesh. The middle box represents the middle 50% of farm prices falling within the interquartile range, which is close to the median in Canada, China, and the U.S. However, 75% of the data fall in the third quartile for Bangladesh.

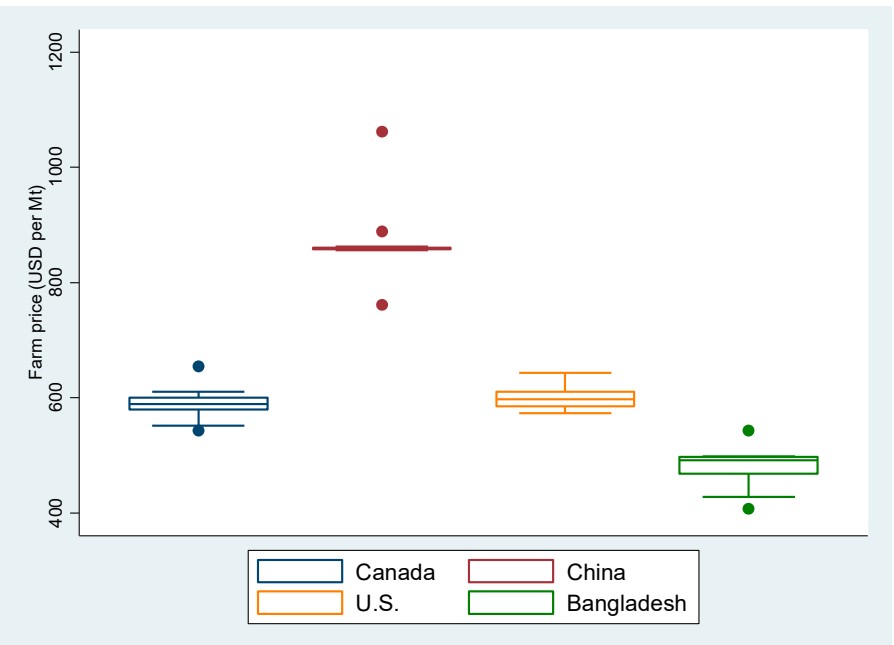

**Figure 3.** Visualization of farm price fluctuations with the box plot.

The lagged average rainfall had a positive consequence in Canada (June) and China (November). Although the lagged rainfall in January and March had a positive effect on the production area in the U.S., a negative consequence was found in May, which might ultimately affect productivity and yield. In Bangladesh, the average lagged rainfall in October had a positive impact on rapeseed during planting time. Most of the applied variables in the yield and area functions were found to be statistically significant (Table 3). Figure 4 represents the area allocation status during the baseline and simulation periods.

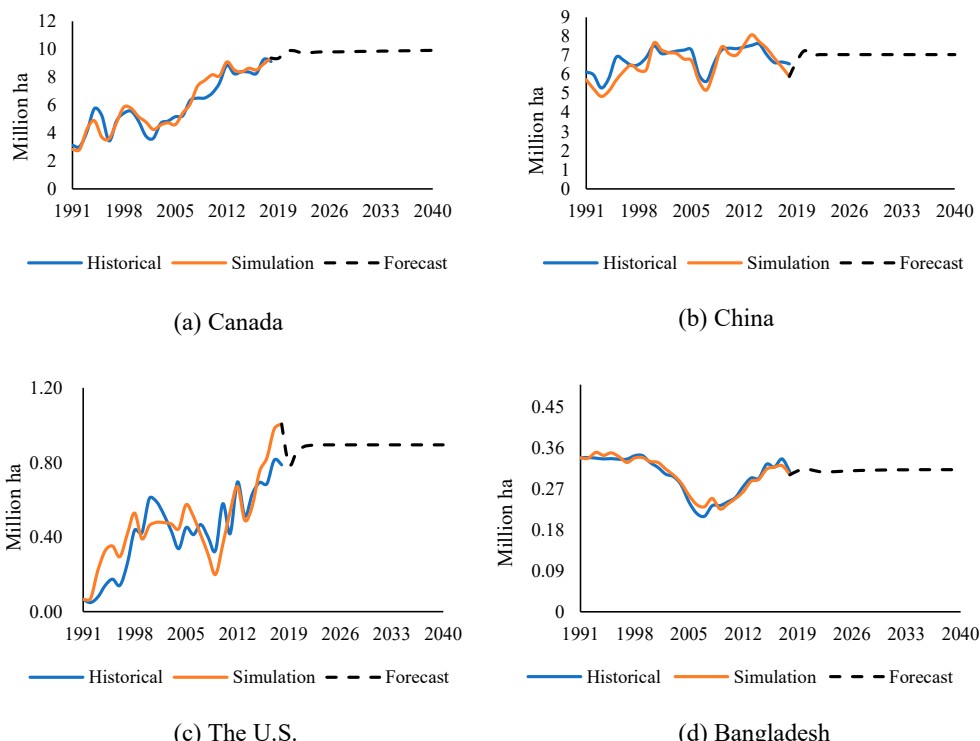

**Figure 4.** Planted area during baseline (1991–2018) and simulation period (2019–2040).

As the estimation period was 1991 to 2018, during the simulation period we assumed the following assumptions followed by Ishiwata and Furuya, 2020; and Jannat et al., 2021.

(1) The forecasted growth of the *CPI* was the average annual growth between 2015 and 2018, (2) the forecasted growth of real GDP and population was the annual average between 2015 and 2018, and (3) the monthly climate variables growth was also calculated between 2015 and 2018. We presented each simulated result for the area, exports, imports, and per capita consumption for the period of 2019 to 2040 in Table A2.

Table 4 shows the trading information parameters for Canada as an exporting country and China, the U.S., and Bangladesh as importing countries [61]. Rapeseed exports from Canada are estimated to increase by an average of 81% annually, with a total of 14 million MT in 2040 (Figure 5). Rapeseed imports in China are estimated to be 1.2 and 1.6 million MT in 2018 and 2040, respectively, while they are 1.5 and 1.7 million MT in the U.S. during the simulation period (Figure 5). The FAOSTAT, 2022 supports this projection [62]. For Bangladesh, imports decreased drastically and were estimated to be 120 thousand MT to 100 thousand MT. The cultivation area is increasing in Bangladesh, which is supported by Miah et al. (2015) and BBS (2019) [63,64]. The findings showed that one thousand MT of total production increase led to an increase in exports of 274 MT in Canada. However, it is revealed that one thousand MT of total production increase leads to a decrease in imports of 335 MT and 901 MT in the U.S. and Bangladesh, respectively (Table 4). Similarly, an increase in the world price of rapeseed in USD reduced oil imports by 137 and 648 MT for China and the U.S., respectively, but for Bangladesh, the total imports decreased by 2.2 MT because of the increased world price. The stock change could be explained by the change in production. The independent variable in the stock change function validated the general postulation and was estimated to have statistical significance for Canada.

Processed rapeseed is used as an edible oil in amounts of approximately 40%, 44%, 42%, and 33% in Canada, China, the U.S., and Bangladesh, respectively [61]. The rest of the amount is used as biodiesel and cake feed for animals. A US dollar increase in the crude world price of rapeseed raised biodiesel demand by 110, 617, and 4429 MT for Canada, China, and the U.S., respectively (Table 5). The per capita oil consumption of rapeseed is projected to increase from 16 to 24 kg year$^{-1}$ between 2018 and 2040 in Canada (Figure 6).

For China, the U.S., and Bangladesh, oil per capita consumption shows an increasing but steady trend, as shown in Figure 6. This might occur because of the competitiveness between the substitutes for rapeseed. The substitute price elasticity of soybeans indicated that a 10% rise in soybean prices could possibly increase rapeseed consumption by 2.3% in Canada. For the U.S., the price elasticity of palm oil indicated that a 10% rise in palm oil prices could possibly increase rapeseed consumption by 1.8% (Table 5).

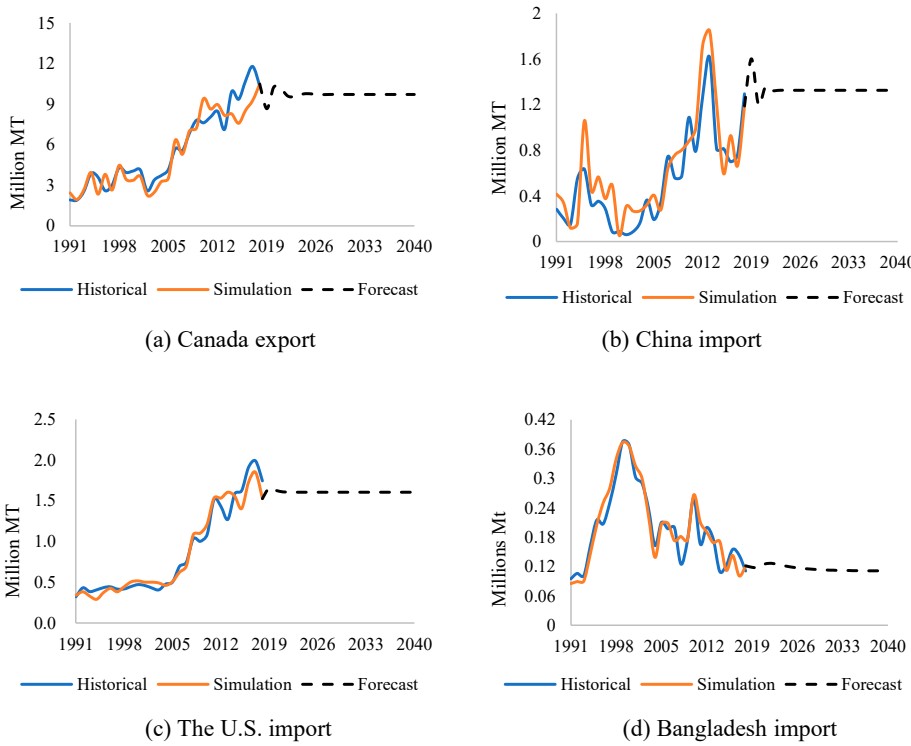

**Figure 5.** Exports and imports status during baseline (1991–2018) and simulation period (2019–2040).

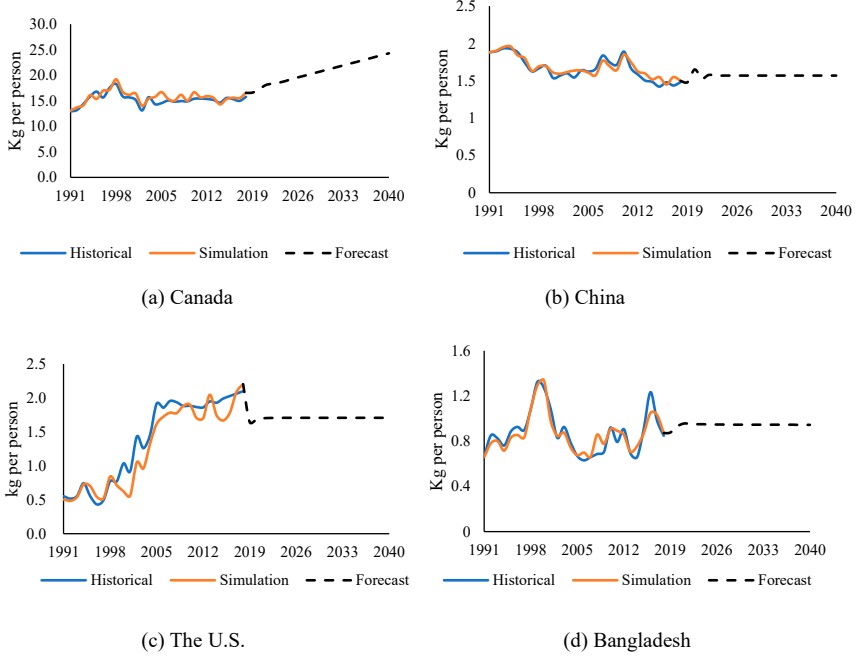

**Figure 6.** Per capita rapeseed oil consumption.

**Table 4.** Estimation of rapeseed exports, imports, and stock change function.

| Parameters | Canada | China | U.S. | Bangladesh |
|---|---|---|---|---|
| | | Exports | | |
| $Q$ | 0.274 **<br>(1.9) | | | |
| $FPd$ | −6576.018 **<br>(−2.21) | | | |
| Adj. R$^2$ (DW) | 0.60 (1.83) | | | |
| | | Imports | | |
| $Q$ | - | - | −0.335<br>(−1.91) | −0.901 **<br>(−2.36) |
| $WPd$ | | | | −2.254 ***<br>(−1.66) |
| Adj. R$^2$ (DW) | - | | 0.60 (2.31) | 0.64 (2.59) |
| | | Stock change | | |
| $dQ$ | 0.411 **<br>(2.35) | | | |
| Adj. R$^2$ (DW) | 0.68 (2.43) | | | |
| | | Oil exports | | |
| $OilQ$ | 0.695 *<br>(7.83) | | | |
| Adj. R$^2$ (DW) | 0.78 (2.05) | | | |
| | | Oil imports | | |
| $OilQ$ | | | −0.717 **<br>(−2.85) | |
| $WPd$ | - | −137.789 **<br>(−1.97) | −648.135 **<br>(−2.29) | |
| $FPd$ | - | 316.605 *<br>(3.43) | | |
| Adj. R$^2$ (DW) | | 0.84 (1.70) | 0.70 (2.38) | |

Source: Authors' estimation, 2021. Note: ***, **, and *, respectively represent levels of significance at 1, 5, and 10%. Values in () denote *t*-values. AdjR$^2$ is adjusted R$^2$ and DW stands for the Durbin–Watson values.

**Table 5.** Estimation of rapeseed biodiesel, per capita oil consumption and price linkage's function.

| Parameters | Canada | China | U.S. | Bangladesh |
|---|---|---|---|---|
| | | Oil biodiesel | | |
| $WPdCru$ | 110.509 **<br>(2.1) | 617.328 **<br>(2.1) | 4429.298 **<br>(2.72) | |
| Adj. R$^2$ (DW) | 0.67 (2.37) | 0.75 (2.25) | 0.56 (2.16) | |
| | | Per capita oil consumption | | |
| $FPd$ | −0.008 **<br>(−2.88)<br>[−0.20] | −0.005 ***<br>(1.69)<br>[−0.19] | −0.002 *<br>(−3.18)<br>[−0.53] | −0.00005 ***<br>(−1.73)<br>[−0.23] |
| $rGDPPC$ | −1.116 *<br>(−4.51)<br>[−4.38] | −0.041 **<br>(−2.76)<br>[−0.63] | −0.099 **<br>(−1.89)<br>[−3.35] | −0.0363 ***<br>(−1.85)<br>[−3.01] |

**Table 5.** *Cont*.

| Parameters | Canada | China | U.S. | Bangladesh |
|---|---|---|---|---|
| *FPdsoy* | 0.012 * (3.56) [0.22] | | | |
| *FPdpal* | | | 0.0004 ** (2.45) [0.18] | |
| Adj. R$^2$ (DW) | 0.63 (2.47) | 0.80 (2.00) | 0.71 (2.23) | 0.67 (2.51) |
| | | Price linkages | | |
| *FP* | - | 0.042 ** (2.18) | 0.244 ** (2.17) | 0.093 ** (3.67) |
| Adj. R$^2$ (DW) | | 0.60 (2.02) | 0.89 (2.04) | 0.68 (2.26) |
| | | Biodiesel price linkages | | |
| *WPcru* | 0.856 * (3.81) | 0.063 ** (1.93) | 0.368 * (3.02) | |
| Adj. R$^2$ (DW) | 0.71 (2.58) | 0.75 (2.05) | 0.75 (1.88) | |

Source: Authors' estimation, 2021. Note: ***, **, and *, respectively represent levels of significance at 1, 5, and 10%. Values in () and [] denote *t*-values and elasticity. AdjR2 is adjusted R2 and DW stands for the Durbin–Watson values.

In addition, the per capita oil consumption (kg per year) for all countries was negatively related to the farm price of rapeseed and GDP per capita. The income elasticity of demand was −4.38, −0.63, −3.35, and −3.01 for Canada, China, the U.S., and Bangladesh, respectively. These results indicate that with an increase in income, demand for rapeseed could be referred to as an inferior good. The price linkage results show that a 10% increase in world prices would increase farm prices by 4.2%, 24.0%, and 9.3% in China, the U.S., and Bangladesh, respectively.

### 3.2. Projection Scenario

Rapeseed Supply and Demand Scenario under Different RCPs and SSPs

The simulation results for rapeseed production for various countries were obtained using model equilibrators. The socioeconomic scenarios were investigated under RCP 2.6, RCP 4.5, RCP 6.0, and RCP 8.5. The equilibrator is a method for finding equilibrium quantities and prices of a supply and demand model in a Microsoft Excel © spreadsheet. This method was developed by the Food and Agricultural Policy Research Institute at the University of Missouri (FAPRI-MU) [65]. The following simulation assumptions were made for this study: (i) The estimated parameters were fixed. (ii) The climatic variables directly affect yields and area. (iii) The average growth of the GDP deflator and exchange rate is assumed between 2015 and 2018. (iv) Each climate scenario exhibits individualistic characteristics that may differ depending on the price and supply scenario, and we also presented the overall projection scenario for rapeseed production (million MT), farm prices (USD MT$^{-1}$), and per capita consumption rates (kg year$^{-1}$ person$^{-1}$) under RCP 2.6, RCP 4.5, RCP 6.0, and RCP 8.5 with corresponding SSPs.

The farm price of rapeseed was estimated as a market-clearing price using the convergence mechanism of supply and demand for the projected period (2019–2060). The farm prices of Canada and the U.S. fluctuated significantly under RCPs, but had an increasing trend over the projected period considering the constant price of 2010 in USD (exchange rate, 2010) (Figures 7 and 8) [66]. This finding also predicted that the rapeseed price would be higher under SSP1 and SSP2 under all RCP scenarios. This indicates that rising national income and edible oil consumption diversification would bring rapeseed consumption to a situation where it increases.

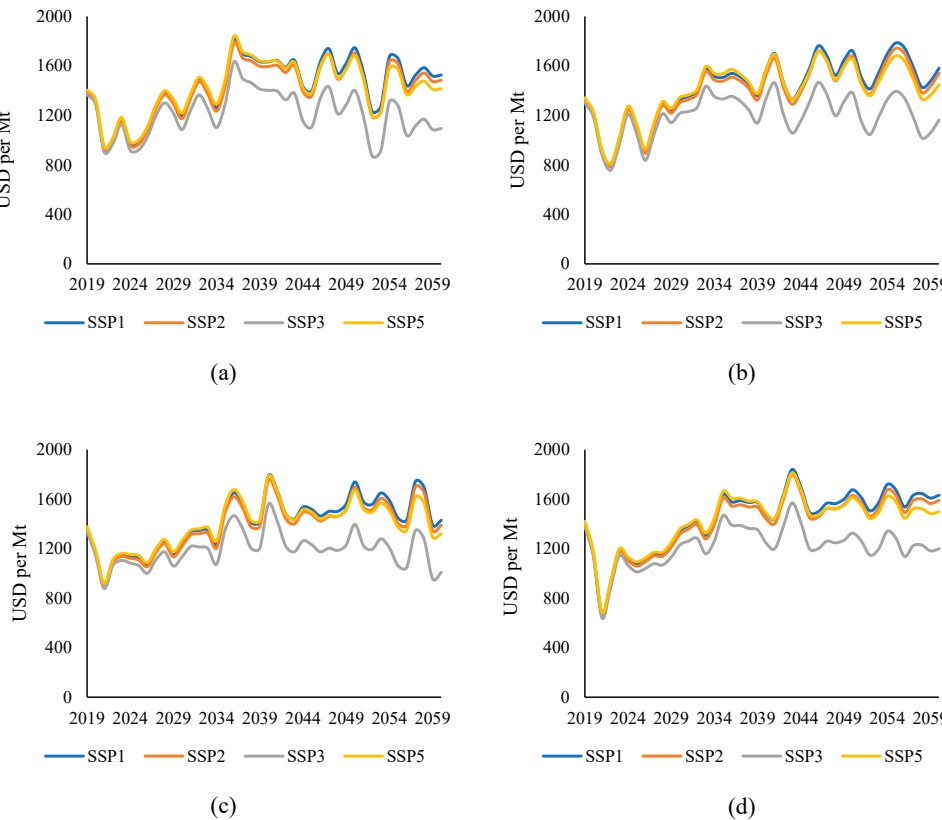

**Figure 7.** Farm price of Canada under (**a**) RCP 2.6, (**b**) RCP 4.5, (**c**) RCP 6.0 and (**d**) RCP 8.5 with different SSP scenarios (2019–2060).

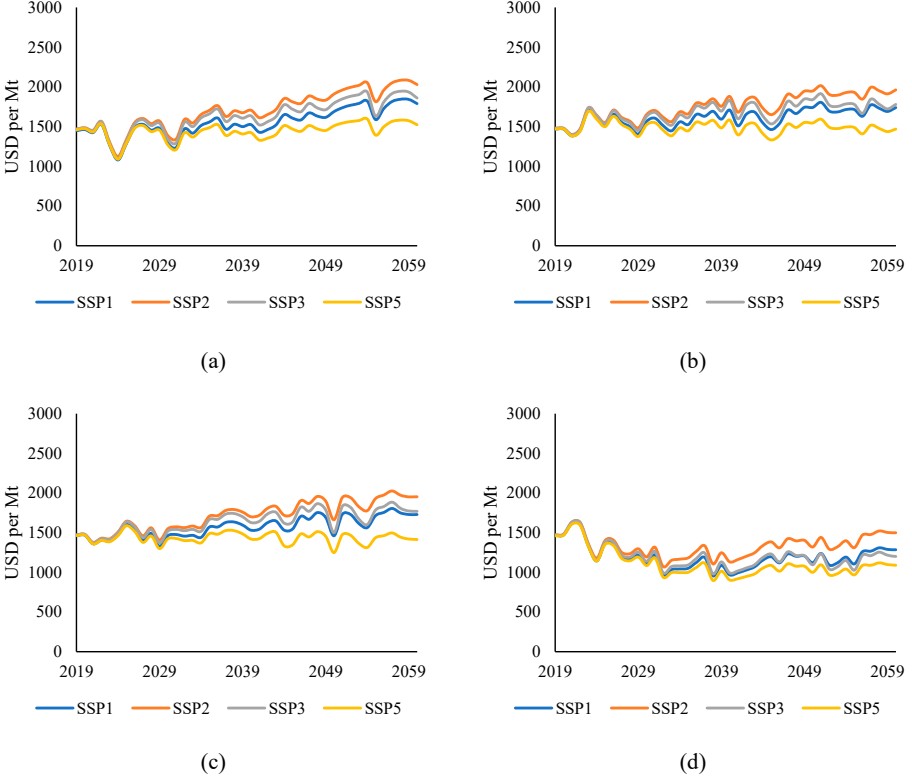

**Figure 8.** Farm price of the U.S. under (**a**) RCP 2.6, (**b**) RCP 4.5, (**c**) RCP 6.0 and (**d**) RCP 8.5 with different SSP scenarios (2019–2060).

The farm price in China fluctuated significantly under the RCPs but had a decreasing trend over the projected period (Figure 9). The supply and demand model's simulation results for China indicated that the effects of climate change influenced rapeseed market price variations. According to the Global Agricultural Information Network (GAIN) report, 2017, the drop is mainly attributed to China's stricter policy on foreign matter (FM) requirements on imported rapeseed. Relatively tight global rapeseed supplies raised rapeseed prices compared to other oilseeds. In addition, the sale of rapeseed oil reserves also discouraged imports during 2015–2016 [67]. The farm price fluctuations were visualized in a box plot under RCP 6.0 and RCP 8.5, with SSP2 as the middle-of-the-road challenges. The box plot illustrates that farm price fluctuation would be more pronounced in RCP 8.5 than in RCP 6.0 since climate variability is greater in RCP 8.5 for Canada, China, and the U.S. (Figure A4).

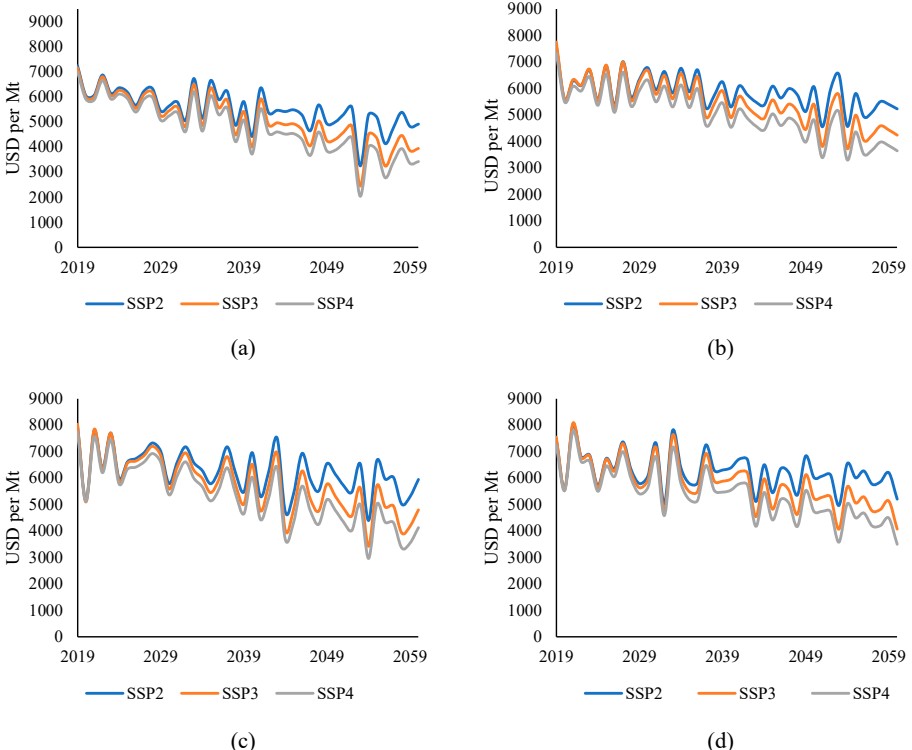

**Figure 9.** Farm price of China under (**a**) RCP 2.6, (**b**) RCP 4.5, (**c**) RCP 6.0 and (**d**) RCP 8.5 with different SSP scenarios (2019–2060).

After obtaining the equilibrium price of rapeseed from the model, we further extended our synthesis by linking how trading countries' farm prices affect the Bangladesh rapeseed market under different scenarios (RCPs and SSPs). The distribution of areas for economic crops is an enormous concern in relation to climate change. From our simulation, we found that the area changes with the potential changes in climate, which can, in turn, impact the production of oilseed rape. Our study observed that there were significant differences among the various scenarios and climate models, but the production and consumption changes in space were, overall, consistent across these climate predictions. Projection results based on the coefficient of variation (CV) of area, production, per capita consumption and farm price of rapeseed are given in Table 6.

The current model estimates that the rapeseed area during the baseline (1991 to 2018) period has been changed by an average of 299.9 ha (10.3%), where the projection up to 2060 shows that it would be changed by 1.1–4.4% under different RCPs with SSP2, SSP3, and SSP4 (Table 6). Fluctuations in rainfall amount in October and February were observed to be slightly higher in RCP 8.5 than in RCP 2.6, RCP 4.5, and RCP 6.0, which would influence

more variation in rapeseed production. The climate variables in October, December, and February would be very important for rapeseed because the planting time is October, as it would start to develop pods in early December and seeds mature in February. Cool rainwater disrupts the maturity of oilseed rape, hampers harvesting, and shrinks the harvest area during February, so planting in mid-September and harvesting in mid-February will give rapeseed the best chance of survival.

**Table 6.** Coefficient of variation (%) of rapeseed market in Bangladesh.

| Scenario | | Area ('000 ha) | Production ('000 MT) | Per Capita Consumption (Kg Year$^{-1}$ Person$^{-1}$) | Farm Price (USD MT$^{-1}$) |
|---|---|---|---|---|---|
| Changes in the baseline period (1991–2018) | | 299.9 (10.3) | 252.3 (53.2) | 1.09 (18.6) | 300.44 (3.9) |
| Projection period (2019–2060) | | | | | |
| RCP 2.6 | SSP2 | 282.3 (3.3) | 336.1 (6.2) | 3.5 (19.6) | 439.16 (4.6) |
| | SSP3 | 281.5 (1.8) | 335.2 (5.2) | 3.7 (21.3) | 412.45 (5.1) |
| | SSP4 | 277939.0 (1.2) | 330.9 (3.6) | 3.8 (22.0) | 445.36 (4.5) |
| RCP 4.5 | SSP2 | 286.4 (2.8) | 342.5 (4.4) | 3.5 (19.0) | 469.13 (4.4) |
| | SSP3 | 285702.6 (1.1) | 341.7 (6.1) | 3.7 (21.9) | 444.95 (4.9) |
| | SSP4 | 281.6 (1.6) | 335967.5 (2.5) | 3.8 (22.9) | 458.23 (4.5) |
| RCP 6.0 | SSP2 | 291.9 (2.1) | 346.7 (5.0) | 3.4 (20.0) | 471.34 (4.7) |
| | SSP3 | 292.0 (4.4) | 346.7 (7.2) | 3.7 (21.8) | 470.12 (5.1) |
| | SSP4 | 286.9 (1.3) | 340.7 (3.1) | 3.8 (23.0) | 548.13 (4.9) |
| RCP 8.5 | SSP2 | 281.6 (3.2) | 338.5 (9.6) | 3.5 (20.9) | 401.57 (4.9) |
| | SSP3 | 278.4 (3.9) | 334.7 (10.4) | 3.7 (21.9) | 409.12 (5.2) |
| | SSP4 | 273.4 (1.4) | 328.6 (6.6) | 3.8 (24.1) | 394.41 (5.0) |

Source: Authors' calculation, 2021. Note: Figures in the parenthesis indicate % change.

In the middle of the century, the average increase in the total production potential under the 12 scenario combinations will be 3.4 million MT (Table 6). However, the rate of change in production is slower than the demand for rapeseed. There will be a significant possibility of increasing rapeseed consumption, and within 2060, rapeseed oil demand will increase by an average of 1.09–3.49 kg year$^{-1}$ person$^{-1}$. Fluctuation of price would also be found to be relatively higher in RCP 8.5 and RCP 6.0 with SSP2 (5.2% and 5.1%) than that in both RCP 2.6 and RCP 4.5 (Table 6). The high volatility of price negatively affects the consumption of low-earning people and producer decisions in the prediction period.

## 4. Discussion

Oilseed crops have been grown for thousands of years as sources of edible and nonedible (industrial) oils for a wide range of end uses, including fuels and bioproducts.

Rapeseed oilseed is currently the third-largest source of vegetable oil globally after palm and soybean [68]. Rapeseed is used in Canada, China, the U.S., India, and most European countries as a source of vegetable oil for edible and nonedible oil purposes [69]. Interest in Brassica (canola/rapeseed/mustard) oilseed crop species for edible and industrial oils has been due to their high oil content and high-protein meal left over after oil extraction because these oilseed species are adapted for production in temperate climatic zones and are able to germinate and grow at low temperatures [68].

Our results indicated that there was a significant effect of climate variables i.e., temperature and rainfall on rapeseed production, which was also supported by Pirjo et al., 2009 and Pullens et al., 2019 [17,18]. As the climate structure changed according to geographical characteristics, the effect also varied from country to country. This study also found that in every crop year, rapeseed production areas increased steadily in Canada, China, the U.S., and Bangladesh. Fridrihsone et al. [28] also discovered that the production of oilseeds has been steadily growing for food, feed, fuel, and industrial applications. Depending on the trend of the rapeseed area under cultivation in Bangladesh, it is possible that imports decreased significantly. Kojima et al. [70] stated that per capita consumption of rape oil increased due to the increasing global demand. Our results showed the overall oil consumption per capita shows an increasing but steady trend for China, the U.S., and Bangladesh. There is a possibility that this will occur due to competition between rapeseed substitutes.

Rapeseed market prices continually respond to multiple influences, such as tight supplies of oilseed and palm oil, including a rise in crude oil prices [71]. With the pace of climate variation and demand, a significant increase in farm prices of rapeseed is visible in different projection scenarios (RCPs and SSPs). However, if government officials and policymakers of Bangladesh take proper steps to identify the nature of price divergence, then it would be possible to sustain growers and satisfy consumers in the long run. Despite the fact that it is a cool crop with a deep root system that prevents soil erosion, produces large amounts of biomass, suppresses weeds, and improves soil tilth with this root system, Bangladeshi policymakers should take measures to promote new and disease-resistant varieties and manage residuals for soil conservation and fertility as other studied countries do to sustain the economy [16].

The study has some limitations that should be mentioned and could be addressed in future studies. Monthly climatic factors (such as rainfall and temperature) are used in the study, but extreme weather events, fertilizer effects, and $CO_2$ concentrations are not taken into account due to data availability, and must be examined in future research. Further research could be conducted in the future to examine climate effect scenarios for other countries since this study only considered major trading countries. In addition, climate models can be improved in the future.

## 5. Conclusions and Policy Implications

Through the application of a supply–demand model, our study aimed to examine the impact of climate variability on rapeseed production and market linkages between net exporters and importers. We used temperature and rainfall as climate indicators, which greatly affected rapeseed production in all the studied countries.

Firstly, as a result of the climatic parameters, it was found that changes in temperature have the largest impact on rapeseed yield, and are positively related to the growing seasons, but negatively related to the maturity stages of rapeseed. Besides this, our simulation explored the uncertainty of the impact of climate change on rapeseed production increases during our simulated period. There will be a significant possibility of a steady increase in rapeseed production, but a sharp increase in consumption within 2060 under different RCPs and SSPs for all studied countries investigated.

Secondly, since the effect of climate change is obvious, so it could potentially affect the farm prices and consumption of net exporting and importing countries which may, in turn, ultimately affect the rapeseed market of developing countries such as Bangladesh. Therefore, to reduce price variations of rapeseed due to climate impacts, the concerned

authorities should create support price policies and timely market information to encourage rapeseed growers.

Lastly, as in other developed countries, relevant agricultural extension organizations (AEOs) and the Ministry of Agriculture (MoA) of Bangladesh should implement advanced technologies along with improved cultivars of rapeseed to stabilize domestic supply and demand.

**Author Contributions:** A.J. and J.F. conceptualized the idea; A.J., Y.I.-I. and J.F. all have contributions regarding the use of software; A.J. conducted formal analysis; A.J. prepared the original draft; A.J., Y.I.-I. and J.F. reviewed and edited the document; and J.F. supervised the research. The submitted manuscript version has been read and approved by all authors. All authors have read and agreed to the published version of the manuscript.

**Funding:** This research received no external funding.

**Institutional Review Board Statement:** Not applicable.

**Informed Consent Statement:** Not applicable.

**Data Availability Statement:** Historical data for rapeseed is available at https://www.fao.org/faostat/en/#data, accessed on 11 January 2022. Climate data are available at http://bmd.wowspace.org/team/homex.php and https://www.ncei.noaa.gov/access/search/data-search/daily-summaries, accessed on 11 January 2022.

**Acknowledgments:** We appreciate the data provided by the BBS, BMD, BARC, FAO, WB, IMF, IPCC, NOAA, EEA, JAMSTEC, and IIASA. The authors are grateful to the University of Tsukuba in Japan for allowing them to conduct their research in their lab. We would also like to thank the More Jobs Better Lives (MJBL) Foundation in Japan for providing scholarships for us to conduct this research there. Motoki Nishimori, Principal Researcher, NARO's Institute for Agro-Environmental Sciences, Japan, provided historical and anticipated climatic data for RCPs and SSPs, which we sincerely acknowledge.

**Conflicts of Interest:** The authors declare that they have no conflict of interest.

## Appendix A

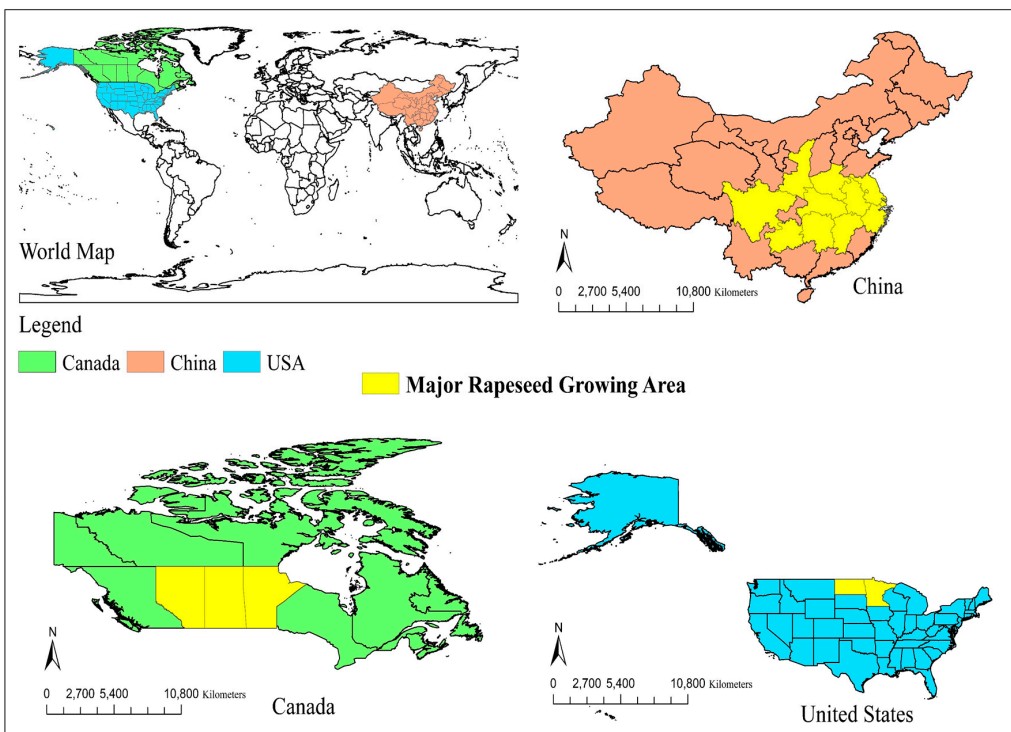

**Figure A1.** Major rapeseed exporting (Canada) and importing (China and the U.S.) countries.

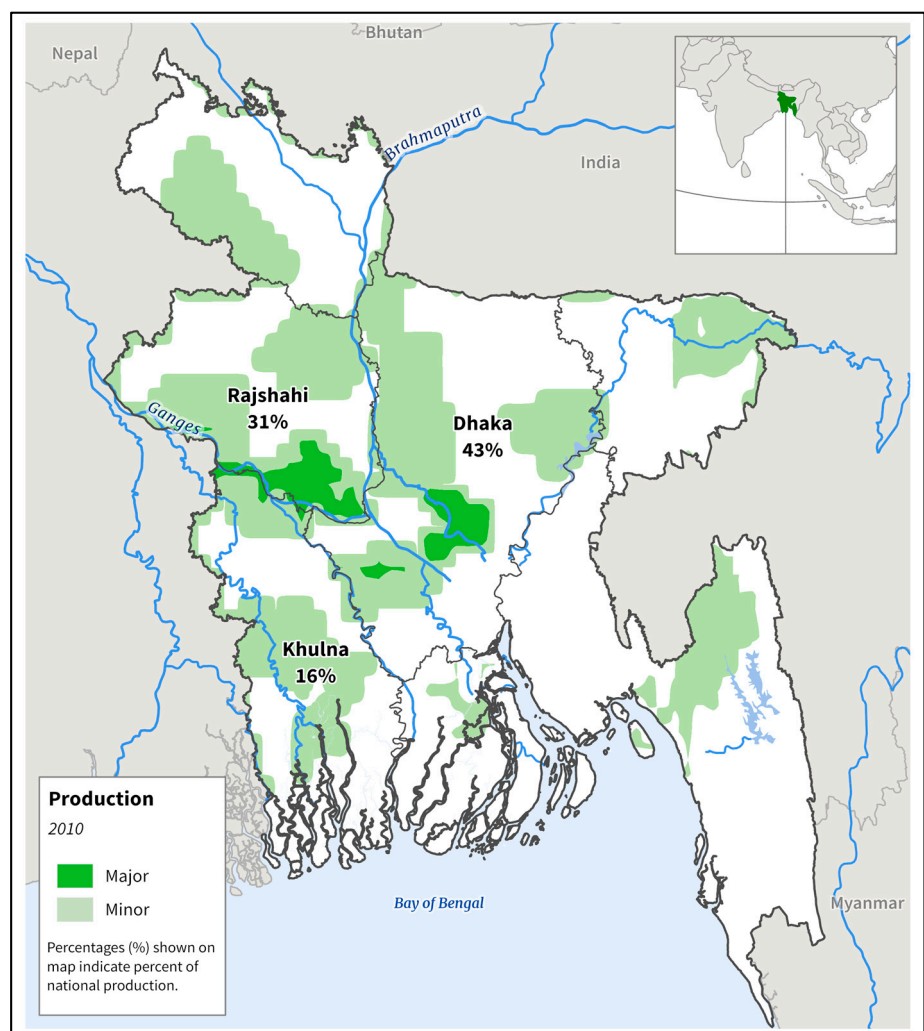

**Figure A2.** Rapeseed Production in Bangladesh showing major and minor growing areas. Source: Adapted with the permission from International Food Policy Research Institute, Spatial Production Allocation Model 2010 v2, 2022, USDA Foreign Agricultural Services. (Available online: https://ipad.fas.usda.gov/countrysummary/images/BG/cropprod/Bangladesh_Rapeseed.png, accessed on 11 January 2022) [72].

| Months | Jan | Feb | Mar | Apr | May | Jun | Jul | Aug | Sep | Oct | Nov | Dec |
|---|---|---|---|---|---|---|---|---|---|---|---|---|
| Canada | | | | | Planting | Planting | Growing | Growing | Harvesting | Harvesting | | |
| China | Growing | Growing | Growing | Harvesting | Harvesting | | | | Planting | Growing | Growing | |
| USA | Growing | Growing | Growing | Growing | Growing | Harvesting | Harvesting | | | Planting | Growing | Growing |
| Bangladesh | Growing | Growing | Harvesting | | | | | | | | Planting | Growing |

Planting (green)     Growing (yellow)     Harvesting (orange)

**Figure A3.** Cropping calendar. Source: Adopted from International Production Assessment Division, USDA and modified by authors. Source: Adapted with the permission from International Food Policy Research Institute, Rapeseed Explorer, 2022, USDA Foreign Agricultural Services. (Available online: https://ipad.fas.usda.gov/cropexplorer/cropview/commodityView.aspx?cropid=2226000&sel_year=2020&rankby=Production, accessed on 11 January 2022) [73].

**Table A1.** Unit root tests for climate variables.

| Variables | Levels | | 1st Differences | | Variables | Levels | | 1st Differences | |
|---|---|---|---|---|---|---|---|---|---|
| | ADF | PP | ADF | PP | | ADF | PP | ADF | PP |
| | | | | | Canada | | | | |
| MayT | –3.84 ** | –3.81 ** | –3.91 ** | –23.36 *** | MayR | –5.68 *** | –6.68 *** | –4.61 *** | –23.40 *** |
| JunT | –3.73 ** | –3.63 ** | –5.19 *** | –19.22 *** | JunR | –5.74 *** | –7.99 *** | –5.82 *** | –16.12 *** |
| JulT | –4.01 *** | –9.83 *** | –4.84 *** | –21.30 *** | JulR | –4.69 *** | –5.12 *** | –6.59 *** | –17.01 *** |
| AugT | –4.68 *** | –4.67 *** | –4.50 *** | –9.38 *** | AugR | –6.20 *** | –6.96 *** | –4.36 *** | –12.19 *** |
| SepT | –4.38 ** | –4.87 *** | –5.89 *** | –11.08 *** | SepR | –4.81 *** | –4.81 *** | –7.40 *** | –16.16 *** |
| OctT | –5.25 *** | –13.34 *** | –4.63 *** | –17.62 *** | OctR | –5.38 *** | –5.66 *** | –8.16 *** | –14.50 *** |
| | | | | | China | | | | |
| JanT | –2.35 | –5.73 *** | –2.60 | –18.99 *** | JanR | –6.88 *** | –6.83 *** | –6.74 *** | –39.38 *** |
| FebT | –6.46 *** | –6.78 *** | –4.99 *** | –24.69 *** | FebR | –4.16 ** | –4.17 ** | –5.78 *** | –19.71 *** |
| MarT | –3.42 * | –3.40 * | –3.63 * | –9.95 *** | MarR | –4.38 *** | –4.40 *** | –4.29 ** | –9.06 *** |
| AprT | –4.01 ** | –4.40 *** | –4.81 *** | –17.32 *** | AprR | –5.28 *** | –5.35 *** | –5.86 *** | –17.87 *** |
| MayT | –3.40 * | –6.92 *** | –4.37 ** | –18.23 *** | MayR | –6.43 *** | –7.38 *** | –5.90 *** | –23.14 *** |
| NovT | –4.94 *** | –5.40 *** | –7.09 *** | –15.88 *** | NovR | –9.58 *** | –11.72 *** | –4.47 *** | –15.36 *** |
| DecT | –4.30 ** | –6.15 *** | –3.63 * | –30.74 *** | DecR | –5.45 *** | –6.35 *** | –6.82 *** | –19.97 *** |
| | | | | | U.S. | | | | |
| JanT | –3.10 | –6.97 *** | –5.26 *** | –43.73 *** | JanR | –5.10 *** | –5.69 *** | –5.14 *** | –20.38 *** |
| FebT | –5.81 *** | –6.04 *** | –7.44 *** | –20.11 *** | FebR | –3.12 | –7.03 *** | –5.32 *** | –15.31 *** |
| MarT | –5.83 *** | –12.65 *** | –5.36 *** | –26.72 *** | MarR | –4.24 *** | –8.38 *** | –4.52 *** | –25.52 *** |
| AprT | –2.20 | –6.70 *** | –8.06 *** | –29.94 *** | AprR | –7.48 *** | –7.47 *** | –4.24 *** | –26.84 *** |
| MayT | –6.61 *** | –6.61 *** | –6.73 *** | –46.68 *** | MayR | –6.94 *** | –6.95 *** | –4.25 *** | –45.91 *** |
| JunT | –2.54 | –5.83 *** | –3.33 * | –18.47 *** | JunR | –6.31 *** | –9.91 *** | –6.14 *** | –43.85 *** |
| JulT | –4.69 *** | –4.22 *** | –3.28 * | –11.19 *** | JulR | –7.19 *** | –7.19 *** | –5.45 *** | –26.21 *** |
| OctT | –5.72 *** | –6.56 *** | –3.14 | –36.33 *** | OctR | –4.74 *** | –14.22 *** | –4.76 *** | –31.37 *** |
| NovT | –5.27 *** | –6.52 *** | –6.41 *** | –32.34 *** | NovR | –1.72 | –10.12 *** | –5.16 *** | –48.73 *** |
| DecT | –5.67 *** | –7.53 *** | –8.21 *** | –29.44 *** | DecR | –2.73 | –6.61 *** | –10.83 *** | –11.88 *** |
| | | | | | Bangladesh | | | | |
| JanT | –6.69 *** | –8.45 *** | –6.70 *** | –24.19 *** | JanR | –7.72 *** | –15.01 *** | –5.64 *** | –26.05 *** |
| FebT | –5.17 *** | –5.61 *** | –7.36 *** | –14.11 *** | FebR | –4.64 *** | –4.64 *** | –8.16 *** | –15.16 *** |
| NovT | –4.30 ** | –4.38 *** | –5.60 *** | –11.97 *** | NovR | –5.49 *** | –12.11 *** | –4.25 ** | –20.06 *** |
| DecT | –4.08 ** | –4.07 *** | –5.91 *** | –12.20 *** | DecR | –5.26 *** | –5.58 *** | –4.88 *** | –11.25 *** |

Note: All the unit root tests include both a constant and a linear trend. ***, **, and * denote significance at 1%, 5%, and 10% levels, respectively. Source: Authors' estimation, 2021.

**Table A2.** Simulated data for area, exports, imports and per capita consumption for the period of 2019 to 2040.

| Year | Area (Million ha) | | | | Exports (Million MT) | | | | Imports (Million MT) | | | | Per Capita Consumption (Kg Year$^{-1}$ Person$^{-1}$) | | | |
|---|---|---|---|---|---|---|---|---|---|---|---|---|---|---|---|---|
| | Canada | China | U.S. | Bangladesh | Canada | China | U.S. | Bangladesh | Canada | China | U.S. | Bangladesh | Canada | China | U.S. | Bangladesh |
| 2019 | 9.3552 | 6.6957 | 0.7881 | 0.3121 | 8.6643 | 1.6020 | 0.8053 | 0.1186 | 16.6179 | 1.4833 | 1.6426 | 0.8771 |
| 2020 | 9.8373 | 7.2325 | 0.8522 | 0.3204 | 10.2614 | 1.2005 | 0.7566 | 0.1183 | 17.2643 | 1.6529 | 1.6777 | 0.9304 |
| 2021 | 9.9102 | 7.0046 | 0.8842 | 0.3187 | 10.1516 | 1.3536 | 0.7731 | 0.1255 | 18.0681 | 1.5313 | 1.6983 | 0.9558 |
| 2022 | 9.7700 | 7.0300 | 0.8922 | 0.3153 | 9.5983 | 1.3231 | 0.7907 | 0.1266 | 18.3601 | 1.5801 | 1.7035 | 0.9529 |
| 2023 | 9.7384 | 7.0333 | 0.8942 | 0.3154 | 9.5645 | 1.3259 | 0.7951 | 0.1243 | 18.5343 | 1.5683 | 1.7048 | 0.9507 |
| 2024 | 9.7915 | 6.8745 | 0.8947 | 0.3166 | 9.7456 | 1.3258 | 0.7962 | 0.1217 | 18.8648 | 1.5699 | 1.7051 | 0.9493 |
| 2025 | 9.8189 | 6.6061 | 0.8542 | 0.3177 | 9.7799 | 1.3258 | 0.7965 | 0.1194 | 19.2547 | 1.5698 | 1.7052 | 0.9483 |
| 2026 | 9.8146 | 6.9776 | 0.8762 | 0.3186 | 9.7230 | 1.3259 | 0.7965 | 0.1175 | 19.5994 | 1.5456 | 1.7052 | 0.9475 |
| 2027 | 9.8149 | 7.0890 | 0.8902 | 0.3194 | 9.7045 | 1.3258 | 0.7863 | 0.1161 | 19.9183 | 1.5559 | 1.7052 | 0.9469 |
| 2028 | 9.8266 | 7.0226 | 0.8937 | 0.3200 | 9.7216 | 1.3258 | 0.7940 | 0.1150 | 20.2496 | 1.5681 | 1.7052 | 0.9494 |
| 2029 | 9.8380 | 6.9792 | 0.8810 | 0.3204 | 9.7301 | 1.3361 | 0.7959 | 0.1141 | 20.5912 | 1.5599 | 1.7052 | 0.9484 |
| 2030 | 9.8455 | 6.8379 | 0.8750 | 0.3207 | 9.7253 | 1.3698 | 0.7964 | 0.1135 | 20.9299 | 1.5728 | 1.7052 | 0.9476 |
| 2031 | 9.8525 | 6.8194 | 0.8735 | 0.3210 | 9.7219 | 1.3008 | 0.7931 | 0.1130 | 21.2648 | 1.5694 | 1.7052 | 0.9479 |
| 2032 | 9.8609 | 6.8909 | 0.8867 | 0.3212 | 9.7349 | 1.3321 | 0.7923 | 0.1126 | 21.6003 | 1.5618 | 1.7052 | 0.9482 |
| 2033 | 9.8694 | 7.0297 | 0.8883 | 0.3214 | 9.7318 | 1.3252 | 0.7921 | 0.1123 | 21.9373 | 1.5571 | 1.7052 | 0.9485 |
| 2034 | 9.8775 | 7.0303 | 0.8833 | 0.3215 | 9.7209 | 1.3259 | 0.7954 | 0.1121 | 22.2742 | 1.5632 | 1.7052 | 0.9480 |
| 2035 | 9.8855 | 6.9466 | 0.8765 | 0.3216 | 9.7207 | 1.3258 | 0.7951 | 0.1119 | 22.6108 | 1.5680 | 1.7052 | 0.9479 |
| 2036 | 9.8935 | 6.8789 | 0.8784 | 0.3216 | 9.7268 | 1.3258 | 0.7939 | 0.1117 | 22.9473 | 1.5736 | 1.7052 | 0.9482 |
| 2037 | 9.9016 | 6.8494 | 0.8829 | 0.3217 | 9.7288 | 1.3284 | 0.7925 | 0.1116 | 23.2841 | 1.5674 | 1.7052 | 0.9482 |
| 2038 | 9.9096 | 6.9133 | 0.8861 | 0.3217 | 9.7270 | 1.3394 | 0.7933 | 0.1116 | 23.6209 | 1.5680 | 1.7052 | 0.9481 |
| 2039 | 9.9175 | 6.9836 | 0.8827 | 0.3218 | 9.7261 | 1.3331 | 0.7942 | 0.1115 | 23.9578 | 1.5628 | 1.7052 | 0.9480 |
| 2040 | 9.9255 | 7.0022 | 0.8794 | 0.3218 | 9.7270 | 1.3347 | 0.7948 | 0.1115 | 24.2947 | 1.5607 | 1.7052 | 0.9481 |

Source: Authors' calculation, 2021.

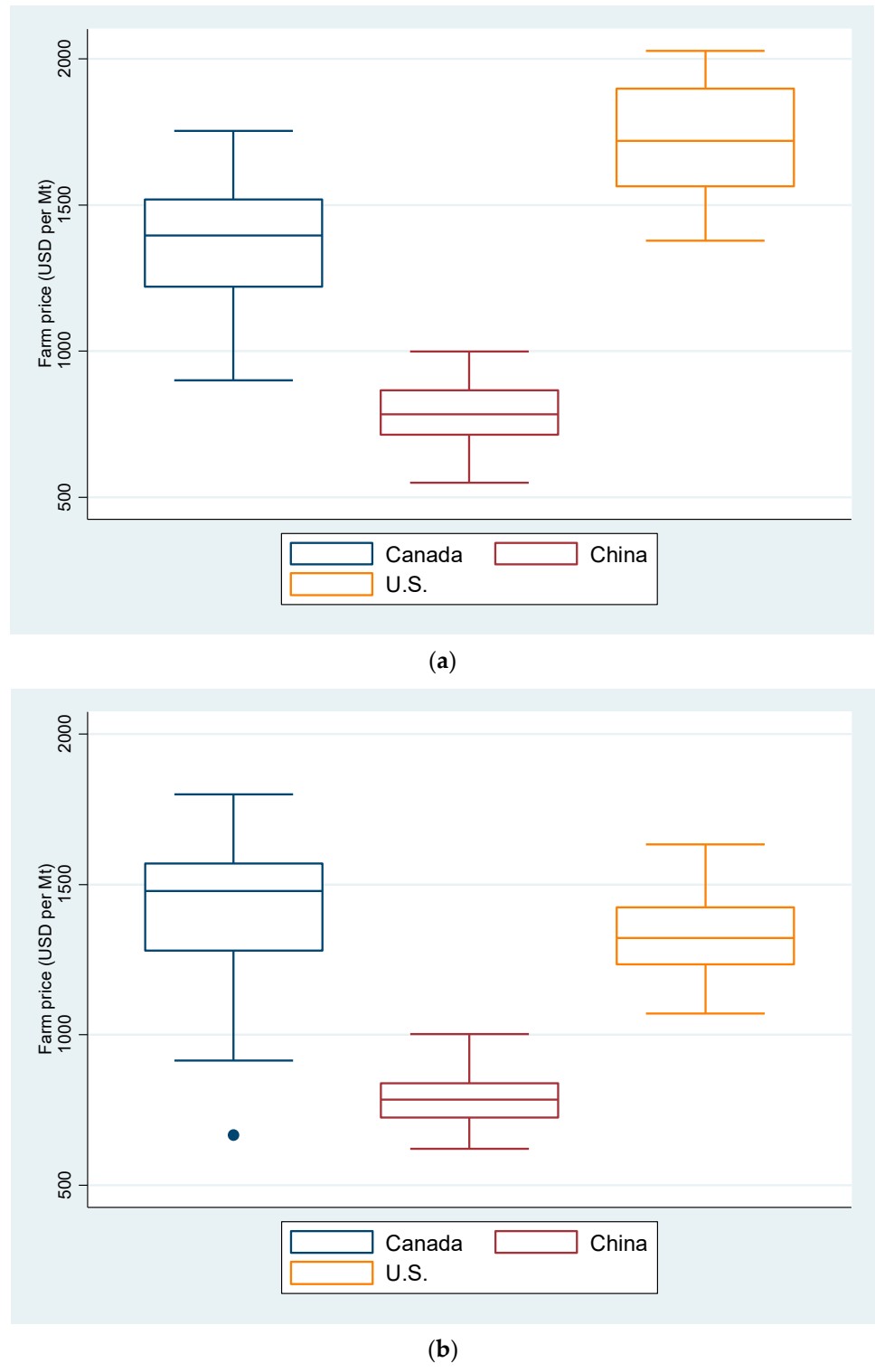

**Figure A4.** Visualization of farm price fluctuations with box plot under (**a**) RCP 6.0 and (**b**) RCP 8.5 with SSP2.

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
