# Peer review of "Does Climate Change Affect Rapeseed Production in Exporting and Importing Countries? Evidence from Market Dynamics Syntheses"

_sustainability, doi:10.3390/su14106051_

Round 1

Reviewer 1 Report

Congratulations for your research!

Reviewer 2 Report

L 32-35: This is unclear

L 47-48: The demand for cereals will increase by 72% but the next lines (L 49-52) state the “need” to cereals will rise by 32%.

L 52: “It has been proven……” Where?

“Introduction” section combines various pieces of information that are not well-related to each other.

There is no mention of simulation in the introduction except when objectives are described.

L 166-167: Why CO2 was not considered as a variable?

L 172-167: If these results were already known, then why this research has been conducted?

L 192: Why 2019. 2020, 2021, periods were simulated when these year already a part of the near past? If the studies were conducted earlier than this period, have the authors compared the real data for these years with their simulations?

Why only four countries were included in this research, and not the other countries important in oilseed production?

Fig 3: recheck the X-axis.

Fig 4: Why the simulated cultivation areas are stagnant? Similar is the case for Fig 5 and Fig 6. I doubt this simulation, however, as I am not an expert on this type of analysis, hence will suggest either editor or some other relevant expert may check this.

L591: What is “seed oil concentration”?

L592-600: Irrelevant to the Discussion; this can maximum be a part of the Introduction.

L 605-609: See the previous comment.

L600-605: This is relevant to the Introduction; the place is just before the objectives i.e., where you explain the knowledge-gap.

L611: Significant effect of what?

L 627: Kg year-1; superscript “-1”.

The Discussion is fragmented and has several irrelevant statements.

L665-666: What is “GM”? Always elaborate all abbreviations at their first use. Further, what is relevance of this statement to the topic of this research?

There could be a shorted and better constructed conclusion.

Reviewer 3 Report

The topic of the paper is interesting as well as the academic contribution of the work. All the scientific sections are well addressed and so I suggest its acceptance after the following minor revisions.

1. Abstract is too long.

2. In the introduction section, the Authors should explain how the article has been structured by presenting the different sections.

3. Tables and figures should report the sources.

4. An extensive editing of English language and style is required.

Reviewer 4 Report

The manuscript entitled “Does Climate Change Affect Rapeseed Production in Exporting and Importing Countries? Evidence from Market Dynamics Syntheses” by Arifa et al. attempted to explore how changes in temperature and rainfall patterns influence rapeseed production and the overall market dynamics of major trading countries. A supply and demand model approach has been employed for major exporting (Canada) and importing countries, i.e., China, the United States (U.S.) along with Bangladesh. The baseline study period was considered from 1991 to 2018, and simulations were performed up to 2040. The findings revealed that the most important effect on rapeseed yield is directly related to changes in temperature, which are positively related to the growing season but negatively related to the maturity stages of rapeseed in all studied countries.

Sufficient analytical data and supply and demand models enable research and results to forecast the future rapeseed oil market and guide the country's policy direction. This research is very practical. A large amount of data and accurate models can provide good evidence for the conclusion. However, there are some points that need attention in order for the article. I would like to recommend a revision is required for the reasons listed below:

1-The Abstract and Introduction sections are too long and more refined language would like to be seen in these sections. 2-In particular, the introduction section is too long to elaborate on the research question. 3-Of course, this research background is very huge, but I think that some research background explanations are unnecessary. 4-Calculations for the significance test of temperature and precipitation and rapeseed yield seem to be lacking in the article. 5-As mentioned in lines 404-406 of the article, several external factors have an impact on rapeseed yield. 6-Therefore, it is necessary to carry out a one-way significance test. 7-It is inappropriate for the purpose of the study to appear in the conclusion. 8-Policy forecasts for Bangladesh appear to be at the heart of the article, however the analysis of the three rapeseed oil importing countries is not much reflected in the conclusions.

Reviewer 5 Report

This paper predicts the impact of climate change on rapeseed production in exporting and importing countries. This topic is interesting, however, several concerns detailed below should be considered.

  1. The abstract must be revised considering the problem, objective, methodology, results, and implications.
  2. More highlighted is highly recommended to be extracted.
  3. The current research gaps and potential future directions must be included.
  4. The results of the simulation should be compared with other studies or the fact in reality to prove the reliability. For example, the author state that “The imports in China are projected to increase by 1.2 and 1.6 million MT between 2018 and 2040, while imports in the U.S. are projected to increase by 1.5 and 1.7 million MT. In Canada, the per capita consumption of rapeseed oil is expected to increase from 16 to 24 kg year-1 person-1 between 2019 and 2040.” The imports in China and the U.S in the year 2018, and the imports in Canada in 2019 are available, the authors should compare their results with the actual imports in these countries.

Round 2

Reviewer 2 Report

The revised version seems okay.

Reviewer 5 Report

The authors have answered all questions in the last review.

This paper can be accepted.